# Marine Biomaterials: Hyaluronan

**DOI:** 10.3390/md21080426

**Published:** 2023-07-27

**Authors:** Rasha M. Abdel-Rahman, A. M. Abdel-Mohsen

**Affiliations:** Institute of Macromolecular Chemistry, Czech Academy of Sciences, Heyrovského Nám. 2, 162 00 Praha, Czech Republic

**Keywords:** marine sources, hyaluronan, extraction process, composites, characterization, wound healing applications

## Abstract

The marine-derived hyaluronic acid and other natural biopolymers offer exciting possibilities in the field of biomaterials, providing sustainable and biocompatible alternatives to synthetic materials. Their unique properties and abundance in marine sources make them valuable resources for various biomedical and industrial applications. Due to high biocompatible features and participation in biological processes related to tissue healing, hyaluronic acid has become widely used in tissue engineering applications, especially in the wound healing process. The present review enlightens marine hyaluronan biomaterial providing its sources, extraction process, structures, chemical modifications, biological properties, and biocidal applications, especially for wound healing/dressing purposes. Meanwhile, we point out the future development of wound healing/dressing based on hyaluronan and its composites and potential challenges.

## 1. Introduction

The vast diversity of organisms living in the oceans presents a wealth of potential for developing high-value bioactive substances and biomaterials [1,2,3]. Although many biomaterials have been derived from marine organisms in recent decades [4,5], there is still a great deal of unexplored potential. Marine biomaterials have unique properties, making them promising materials for various biological and biomedical applications [6,7]. The extraction and processing methods and chemical and biological characteristics of common marine polysaccharides and proteins are important in developing these materials [8,9]. Marine biomaterials have important applications in various fields, including anticancer, antiviral, drug delivery, tissue engineering, and others [10,11,12]. Overall, the potential of marine biomaterials is a rich area of research with promising implications for the future of medicine and materials sciences. Marine biological materials have several advantages over traditional materials, making them attractive for biomedical applications. They are biocompatible and biodegradable and possess unique biological activities that make them suitable for use as scaffold materials, wound dressings, and drug delivery systems for tissue engineering products.

Glycosaminoglycans (GAGs) are a group of complex carbohydrates found in various body tissues, including cartilage, bone, skin, and blood vessels. Also, the marine environment is a rich source of GAG-like molecules that can be used as bioactive components for tissue regeneration [1,13]. These molecules come from various sources, including microorganisms, animals, invertebrates, and seaweeds. One advantage of using marine-derived GAG-like molecules is that they are often structurally distinct from mammalian GAGs, which can provide unique properties and potential therapeutic applications. For example, some marine-derived GAGs have been shown to have anti-inflammatory and antioxidant properties that could be useful in tissue engineering and regenerative medicine [14].

GAGs are involved in the development and differentiation of cells. They are essential for forming tissues and organs during embryonic development [15,16]. GAGs also play a critical role in wound healing and tissue repair. One of the significant advantages of GAGs in tissue engineering is their ability to increase growth factor (GF) bioavailability and reduce their degradation by extracellular proteinases [17]. They stimulate cell proliferation and differentiation, and they are essential for the formation of new tissue. GAGs can bind to GFs and protect them from degradation, increasing their bioavailability and prolonging their activity. This ability to protect GFs from degradation is particularly important in tissue engineering, where the goal is to create an environment that promotes tissue regeneration and repair [18,19,20].

In the skin, hyaluronic acid (HA) is the main GAG found in the dermis, which is the deeper layer of the skin, while chondroitin sulfate and dermatan sulfate are predominantly found in the epidermis, which is the outermost layer of the skin. The distribution and composition of GAGs in the skin vary depending on the location and function of the tissue. During the healing process, there is an increase in the synthesis of hyaluronic acid (HA) in both the epidermis and dermis. This increase in HA production is believed to play a role in forming the extracellular matrix (ECM) and in the recruitment and proliferation of cells involved in wound healing [21,22].

Marine hyaluronic acid (HA) has several advantages that make it a valuable biomaterial for various applications. Some of its key benefits include:Biocompatibility: Hyaluronic acid is naturally present in the human body, particularly in connective tissues, joints, and skin. It has excellent biocompatibility, meaning it is well-tolerated by living tissues and does not typically cause adverse reactions or immune responses when used in medical applications.Water-binding capacity: HA can exceptionally bind and retain water molecules. It can hold up to a thousand times its weight in water, contributing to its lubricating and cushioning properties. This water-binding capacity is crucial for applications such as hydration, moisturization, and lubrication of tissues and joints.Viscoelasticity: HA exhibits viscoelastic behavior, which means it can deform under stress and return to its original shape. This property is beneficial for applications where cushioning and shock absorption are required, such as in joint lubrication or as a component of viscoelastic solutions.Wound healing properties: HA plays a crucial role in wound healing processes. It helps create a moist environment that promotes cell migration, proliferation, and tissue regeneration. HA-based wound dressings can provide a barrier against external contaminants while facilitating wound healing and reducing scarring.Drug delivery capabilities: Hyaluronic acid can be modified to form drug delivery systems. Its high-water content allows for the encapsulation and controlled release of various therapeutic agents, including small molecules, proteins, and nucleic acids. HA-based drug delivery systems can protect sensitive drugs, improve stability, and provide sustained release profiles.Tissue engineering and regenerative medicine: HA hydrogels and scaffolds are widely used in tissue engineering and regenerative medicine. They provide a suitable three-dimensional environment for cell growth, migration, and differentiation. HA-based scaffolds can mimic the natural extracellular matrix and promote tissue regeneration in damaged or diseased tissues.

Biocompatibility, water-binding capacity, viscoelasticity, and other advantageous properties make hyaluronic acid a versatile and promising biomaterial for various applications in medicine, cosmetics, and other industries.

In 2010, the global HA treatment market was foreseen to generate over $13.5 billion in revenues, demonstrating an overall compound annual growth rate (CAGR) in excess of 8.1%. The main sectors of extreme commercial interest are those related to dermal fillers, ophthalmic viscoelastic, dermal fillers viscosupplementation and wound healing dressing materials.

## 2. Chemical Structure of Hyaluronic Acid

Hyaluronan, also known as sodium hyaluronate, is a naturally occurring polysaccharide that plays important roles in many biological processes, including wound healing, tissue repair, and lubrication of joints [23]. It is indeed a unique compound among the glycosaminoglycan (GAG) family. While other GAGs are relatively smaller in size. HA structure (Figure 1) is composed of disaccharide units *D*-glucuronic acid and *D-N*-acetylglucosamine, and are linked together through alternating *β*-1,4 and *β*-1,3 glycosidic bonds [24]. HA is synthesized as a free linear polymer without any protein core, and its molecular weight can range from 2 × 10^5^ to 10^7^ KDa, making it one of the largest molecules in the body [25].

HA is also non-sulfated, which sets it apart from most other GAGs. This lack of sulfate groups means that HA does not have a strong negative charge, making it more hydrophilic and less likely to interact with proteins. HA is found in various tissues and body fluids in vertebrate animals, including humans [26]. It is a major component of the extracellular matrix (ECM) and is also present in synovial fluid, the vitreous humor of the eye, and the umbilical cord [27]. It can also be extracted directly from marine animal sources, such as cartilage, as well as from the vitreous humor of several fish species [28]. Currently, hyaluronan production is performed on a large scale using different methods and sources.

### 2.1. Sources of Marine Hyaluronan

Marine hyaluronan is a type of hyaluronic acid derived from marine sources, including fisheyes, sharks (skin and cartilage), swordfish, zebrafish, mollusk bivalve, liver of stingrays, and shellfish [29,30,31,32]. It is similar in structure and bioactivity to hyaluronan derived from mammalian sources such as rooster combs or human umbilical cords, but there are some differences in its properties. One significant difference is the molecular size of marine hyaluronan, which is believed to be smaller than hyaluronan derived from mammalian sources. This may allow it to penetrate the skin more effectively, improving hydration and plumping [31]. However, other studies have reported that hyaluronan derived from fish sources, such as salmon and trout, may have a higher molecular weight and viscosity compared to hyaluronan derived from mammalian sources [33]. This may be due to differences in production and processing methods and the species-specific properties of hyaluronan. Additionally, some research indicates that marine hyaluronan may have antioxidant and anti-inflammatory properties, which could benefit overall skin health (Figure 2).

Marine-derived hyaluronan has become popular as a cosmetic ingredient in recent years, with many skincare products containing marine hyaluronan to help hydrate and plump up the skin [33]. In addition, to its cosmetic benefits, marine hyaluronan has also been studied for its potential therapeutic applications, such as wound healing, tissue repair, and arthritis treatment. However, more research is needed to fully understand its effects on the body and its potential benefits [34].

### 2.2. Extraction Process of Hyaluronan

Several methods were used to extract hyaluronan from marine sources, including chemical, detergent and enzymatic [35]. The process typically involves first breaking down the cellular matrix of the tissue using a physical or chemical process, such as grinding or homogenization, followed by the use of enzymes or organic solvents or detergents to extract the hyaluronan with different molecular weights with high dispersibility [30].

#### 2.2.1. Chemical Process

The organic solvent method is one of the commonly used methods for extracting hyaluronan from marine sources, and it involves the use of acetone to extract the HA from the organism’s tissue. This method requires several purification steps to obtain high pure HA [35]. The use of cetylpyridium complex (CPC) as a complexing agent is another method for isolating HA, and it is relatively fast and efficient [36,37]. In this method, CPC forms a complex with the HA molecule, allowing it to be separated from other tissue components. The HA-CPC complex is then purified using an anion exchanger such as DEAE-Sephadex A-25, which separates the complex based on charge differences. However, this method requires specialized equipment and expertise. The choice of extraction method will depend on various factors, such as the type of marine source, the desired properties of the final product, and the end-use applications [38].

#### 2.2.2. Enzymatic Treatments

The use of enzymes such as hyaluronidase [39], chondroitinase [40] or papain [41] is a common method for extracting hyaluronan from marine sources. These specific enzymes are used to break down the connective tissue and release hyaluronan. The resulting mixture is then treated with protease enzymes to remove any protein impurities that may be present in the extract. After this, the hyaluronan is purified using precipitation and washing steps similar to those used in chemical methods [42]. This method can be effective but may require additional purification steps to remove any remaining impurities and obtain a pure specific molecular weight hyaluronan product. The use of enzymes in the extraction process can be more specific and gentler than chemical methods, leading to a higher yield of pure hyaluronan. However, it also requires specialized equipment and expertise in handling enzymes [30].

#### 2.2.3. Detergent Method

In this method, the tissue is first homogenized to break down the cellular matrix and then treated with a mixture of detergents such as Triton X-100 and sodium deoxycholate to solubilize the cellular membrane and release the hyaluronan [43]. This extract is then treated with DNase and RNase enzymes to remove nucleic acids and then with protease to remove proteins. Finally, the hyaluronan is precipitated with ethanol or isopropanol and purified by dialysis or size exclusion chromatography [44]. This method can be efficient and yields a pure hyaluronan product, but it may also require additional purification steps to remove any remaining impurities.

On the other hand, the yield of hyaluronan extracted can be varied depending on the source of the tissue or organism (Table 1). Studies have shown that marine sources like fish and shellfish tend to have higher hyaluronan levels than terrestrial sources like cows or pigs [45]. For instance, some types of fish scales and cartilage have been found to contain high levels of hyaluronan, which can be extracted using enzymatic or chemical methods [46]. However, the extraction method and conditions can also affect the amount of hyaluronan that can be obtained and the quality and purity of the final product [47].

### 2.3. Hyaluronan Molecular Weight

The unique properties of HA make it highly versatile in its functions. It is found in almost every tissue in the body, where it helps to provide structural support and hydration to the extracellular matrix. HA exists in various forms, including high molecular weight (HMW) up to 1 MDa, medium molecular weight (MMW) from 500–700 KDa, low molecular weight (LMW) from 50–150 KDa and cross-linked derivatives [21]. The molecular weight of HA is an important factor that can affect its biological properties. For medical applications, low-polydispersity or monodisperse HA is preferred to ensure consistent and predictable biological effects [25].

Monodisperse HA can be prepared by breaking down and reassembling HA chains through successive degradation cycles. Recent studies have challenged the idea that HA of different molecular weights affects the same receptors differently. However, it has been shown that very high molecular weight HA, such as 6000 kDa HA produced by naked mole rats, can suppress the signaling of CD44, leading to altered expression of a subset of p53 target genes [50]. This suggests that HMW-HA may have cytoprotective properties. It’s important to note that differences in the genes regulated by p53 exist between different species, so this investigation is limited to human cells [51]. Overall, the molecular weight of HA is an important area of research that plays a critical role in its biological effects. HMW-HA is found in healthy tissues and has anti-inflammatory properties. It has been shown to inhibit inflammation by preventing cell growth and differentiation, decreasing the production of inflammatory cytokines by various types of cells, and impairing phagocytosis by macrophages. In addition to its anti-inflammatory properties, HMW-HA has also been implicated in inhibiting tumor progression [52].

Recent studies have suggested that the administration of HMW-HA is effective in reducing inflammation in models of lung injury, collagen-induced arthritis, and other in vivo model systems. HMW-HA also plays an important role in maintaining tissue structure and function. For example, it can help maintain the cohesion and structure of the epithelium, lubricate articulations, and play a pivotal role in skin repair [53]. These properties make HMW-HA useful in various medical applications, such as dermal fillers, ophthalmic solutions, and wound dressings [54].

On the other side, LMW-HA has been shown to have pro-inflammatory properties and can stimulate the production of growth factors, which can lead to increased cell proliferation and tissue regeneration. It can also interact with receptors and induce biological pathways, including inflammatory responses [53]. While HMW HA generally inhibits inflammation, LMW HA can have the opposite effect and contribute to the development of inflammation. Therefore, the molecular weight of HA is an essential factor in determining its biological effects and potential medical applications [55].

### 2.4. Properties of Hyaluronan

Hyaluronic acid (HA) is highly biocompatible because it is a naturally occurring polysaccharide found in many body tissues, including the skin, joints, and eyes [56]. HA is generally well-tolerated and non-immunogenic, unlikely to cause an allergic reaction or elicit an immune response when used in medical or cosmetic applications. In addition to its biocompatibility, HA is also biodegradable. It can be broken down by hyaluronidase, an enzyme present in many tissues and responsible for the turnover of hyaluronan in the body [26].

Also, the gut bacteria can degrade hyaluronan, allowing for its natural elimination from the body. The biodegradability of hyaluronan makes it an ideal material for temporary medical devices, such as drug delivery vehicles or tissue engineering scaffolds. These devices can be designed to degrade over time, allowing for the controlled release of therapeutic agents or the gradual replacement of the device with natural tissue [57].

The unique properties of Hyaluronan make it well-suited for promoting cell growth, migration, and tissue regeneration. Using it as a building block, scientists can create various biomaterials to support tissue repair and regeneration. For example, HA hydrogels can be used as injectable matrices to deliver cells or drugs directly to a specific tissue site [58] and HA injections are commonly used in cosmetic procedures to fill wrinkles and add volume to the skin [59].

HA is often used as a carrier for other wound healing agents and in cosmetic formulations because it enhances the delivery and absorption of other active ingredients into the skin. When used as a carrier, HA can form a protective film over the skin that helps to retain moisture and improve the penetration of other ingredients. This can help to improve the effectiveness of other active ingredients, such as peptides, antioxidants, and vitamins, in promoting wound healing or improving skin health and appearance [60].

Furthermore, hyaluronan is highly hydrophilic. It can absorb and retain many water molecules, making it an excellent hydrating agent and a key component in maintaining tissue hydration and lubrication. Due to its water-retaining ability, hyaluronan exhibits a unique viscoelastic behavior similar to natural tissues. This property provides cushioning and shock-absorbing effects in joints and other tissues [13].

The interaction between hyaluronan (HA) and cells plays a significant role in various biological processes and has implications for tissue development, homeostasis, and disease progression. Here are some key aspects of the interaction between HA and cells:Cell adhesion: HA can interact with cell surface receptors, such as CD44 and RHAMM (Receptor for HA-mediated Motility), leading to cell adhesion. This interaction involves cell migration, tissue morphogenesis, and wound healing. CD44, in particular, is a primary cell surface receptor for HA and is expressed in many cell types, including immune cells, stem cells, and cancer cells.Receptor-mediated signaling: The binding of HA to cell surface receptors can initiate intracellular signaling pathways, leading to various cellular responses. These signaling pathways regulate cell proliferation, migration, differentiation, and survival. For example, the interaction between HA and CD44 can activate signaling pathways involving kinases, phosphoinositide 3-kinase (PI3K), and mitogen-activated protein kinases (MAPKs).Cell migration: HA can act as a substrate for cell migration. HA’s high molecular weight and hydrated nature provide a physical framework that allows cells to migrate through tissues. HA-rich matrices can guide and direct cell movement during embryonic development, tissue repair, and immune responses.Cell differentiation and stem cell behavior: HA has been shown to influence cell differentiation and stem cell behavior. HA hydrogels and scaffolds can provide a supportive microenvironment for stem cells, promoting their self-renewal and directing their differentiation into specific cell lineages. HA’s mechanical properties and bioactive cues can influence stem cell fate decisions.Inflammation and immune response: HA plays a role in modulating inflammation and immune responses. HA fragments generated during tissue injury or degradation can activate immune cells and induce pro-inflammatory cytokine production. The interaction between HA and immune cells can regulate immune cell trafficking, activation, and function.Extracellular matrix organization: HA is a major extracellular matrix (ECM) component and contributes to its structural integrity. HA interacts with other ECM molecules, such as collagen and proteoglycans, to form a hydrated and dynamic matrix. This matrix provides mechanical support, regulates tissue architecture, and influences cell behavior and function.

It’s important to note that the specific effects of HA on cells can vary depending on factors such as HA molecular weight, concentration, and the presence of other signaling molecules or growth factors. The interaction between HA and cells is complex and context-dependent, contributing to various physiological and pathological processes in the body.

### 2.5. Chemical Modification of Hyaluronan

Hyaluronic acid (HA) is a versatile molecule that can be modified in numerous ways to tailor its properties and functions to specific needs. Some of the modifications commonly employed in HA include chemical modifications, cross-linking with other molecules or polymers, and the formation of hydrogels, scaffolds, or membranes [54]. Chemical modifications of HA involve introducing functional groups to alter its properties. This can be done through various chemical reactions, such as esterification [61], amidation [62], oxidation [63], or conjugation with other molecules [64] (Figure 3). These modifications can impact the degradation rate, solubility, mechanical properties, and bioactivity of HA, making it suitable for specific applications. Cross-linking of HA involves connecting HA molecules with other molecules or polymers using cross-linking agents, such as divinyl sulfone or carbodiimides, and it can be used to form covalent bonds between HA chains, creating a three-dimensional network [65].

Hyaluronan has different active surface groups like hydroxyl, carboxyl and acetamide groups that could interact easily and generate new compound material with different functionality and properties (Figure 3).

The esterification reaction of the OH groups of hyaluronan is a little problem due to the incomplete solubility of hyaluronan in solvents suitable for acylation reactions and the lower reactivity of the secondary hydroxyl groups (Figure 3). More harsh conditions like high temperature and reaction time are needed with involving the primary OH group only. Hyaluronan methacrylate is a versatile macromer to synthesize functional coating and photopolymerizable materials like scaffolds and hydrogels [65]. The acrylation reaction of hyaluronan can be done by reaction of hyaluronan with methacrylate anhydride under slightly basic conditions and ice cooling temperature (Figure 3). The consistent hyaluronan acrylate is synthesized with acryloyl chloride using CH_2_Cl_2_/H_2_O mixture in the presence of tetra-n-butylammonium fluoride as phase transfer catalyst agent. Another strategy of the esterification of hyaluronan is epoxidation reactions. Glycidyl methacrylate is one of the most common epoxide-end groups used for the esterification of hyaluronan in the presence of the catalytic amount of triethylamine (basic condition) (Figure 3).

Common oxidation reactions of hyaluronan are done using different oxidizing agents [66] like periodate (NaIO_4_), 2,2,6,6-tetramethyl-piperidinyl-1-oxy radical (TEMPO) [67], hydrogen peroxide (H_2_O_2_), and ammonium persulfate (NH_4_)_2_S_2_O_8_). Using a periodate oxidizing agent could change the hyaluronan backbone and lead to cleavages of the sugar ring, forming corresponding carbonyl moieties (di aldehyde) with a low reaction yield percentage. 2,2,6,6-tetramethyl-piperidinyl-1-oxy radical (TEMPO) is one of the best oxidizing agents used for hyaluronan oxidation due to the regioselectivity of oxidation of primary hydroxyl groups and enhances the sugar ring form is left unaffected (Figure 3). The mechanism of oxidation of primary -OH group is well known and discussed in many published works with high yield percentage [65].

The amidation reaction of carboxylic groups of hyaluronan is synthesized using amine, hydrazine or dihydrazone, which are one of the most used methods for amidation of hyaluronan [68]. First carboxylic groups are activated like 1-ethyl-3-[3-(dimethylamino)-propyl]- carbodiimide (EDC), N-hydroxysulfosuccinimide (NHS). Both activation agents interact with the carboxylic groups of HA, and the amine derivatives couple with the COOH of HA. The byproducts of the amidation reaction are acyl urea derivatives which is water soluble. The advantage of using EDC/NHS is considered a zero crosslinker spacer agent [69] (Figure 3).

Partially deacetylation reaction of hyaluronan was done using different deacetylated agents like a high-temperature hydrazine/hydrazine sulfate mixture in the presence of iodic acid [70]. The partially deacylated hyaluronan was also done with sodium hydroxide for a long reaction time [71]. Unfortunately, all deacylated agents show disadvantages like hydrolysis of the HA chain or generating new end-groups (carbonyl or aldehyde) [65] (Figure 3).

## 3. Forms of Hyaluronan

Wounds are injuries that damage the skin, tissues, and/or organs. They can be caused by various factors, such as trauma, surgery, burns, or underlying medical conditions [72,73,74]—growth novelty in wound care and improved wound closure through information on physiological levels of the healing process. Wounds can be divided as acute or chronic, depending on the time for healing. Acute wounds, usually produced by surgery, mechanical trauma, and burns, heal fast without difficulties through typical healing phases. Chronic wounds, including arterial, venous, diabetic and pressure ulcers, suffer to heal through normal and orderly physiological processes and can take up to several decades to heal with serious complications, amputation or even death. Marine hyaluronan-based wound healing is used in three typical wound repair types listed below in Table 2.

Wound healing is an interactive process due to the need to restore the cellular structure and the epidermal tissue as it was before being wounded [90,91]. This process involves multiple stages, including hemostasis, inflammation, proliferation, and remodeling; it can be affected by various factors, such as age, nutrition, and underlying medical conditions. Adequate wound care is essential to promote proper healing and prevent complications such as infection or delayed healing [90] (Figure 4).

The first phase, inflammation, is initiated immediately after injury and is essential for the healing process to begin. During this phase, damaged tissues release signals that activate the immune system and trigger the recruitment of immune cells to the site of injury [92]. Hemostasis, which is the process of stopping bleeding, also occurs during this phase. Platelets have a crucial role in the hemostatic response by forming a plug to stop bleeding and releasing various growth factors, including platelet-derived growth factor (PDGF) [93]. PDGF stimulates the migration and proliferation of various cells, including neutrophils, which are important for cleaning the wound of debris, bacteria, and dead tissue [94]. In addition to neutrophils, other immune cells, such as macrophages, are also recruited to the wound site during inflammation. Macrophages play a critical role in the later phases of the healing process by clearing debris and secreting factors that promote tissue repair. The inflammation phase sets the stage for the subsequent phases of healing by preparing the wound site for the formation of new tissue [95,96,97].

During the proliferation phase, fibroblasts migrate to the wound site, producing and depositing collagen to form a new matrix [98]. Collagen provides the structural framework for new tissue growth and helps to reestablish the tensile strength of the tissue. In the remodeling phase, the newly formed tissue undergoes a process of reorganization and realignment of collagen fibers. This makes the tissue stronger and more resistant to mechanical stress [99,100].

The remodeling phase can take several months to years, resulting in a scar that is stronger and more resilient than the original tissue [101]. It’s important to note that the degree of scarring can vary depending on factors such as the size and depth of the wound (Figure 4).

Bacterial infection is a common complication of wound healing and can significantly slow down or even prevent the healing process [102]. This is because bacteria can cause inflammation, damage new tissue, and delay the formation of new blood vessels. Hyaluronic acid effectively prevents bacterial infections in wounds by forming a protective barrier that inhibits the penetration of bacteria. This barrier helps to keep the wound clean and reduces the risk of infection, allowing the body to focus on the healing process [103]. Also, hyaluronic acid has antimicrobial properties that can directly kill certain bacteria, further reducing the risk of infection [104].

HA is a versatile molecule that contributes to many different aspects of wound healing. During the inflammatory phase, HA helps to promote the migration of immune cells to the wound site and modulates their activity to help control inflammation. In the proliferation phase, HA helps to maintain hydration and provides a scaffold for new cell growth. It also interacts with growth factors and other signaling molecules to promote cell proliferation and differentiation [104]. In the remodeling phase, HA helps regulate collagen synthesis and organization, which is essential for forming a strong scar [105]. It also plays a role in maintaining tissue hydration and elasticity. The interaction of hyaluronan with cell surface receptors such as CD44 plays an essential role in various cellular processes. CD44 is a transmembrane glycoprotein involved in cell adhesion and migration and is highly expressed in many types of cells, including those involved in inflammation and wound healing [106].

Fibroblasts play a critical role in the synthesis and degradation of hyaluronan in the extracellular matrix. They are responsible for producing and secreting high molecular weight hyaluronan into the matrix, where it interacts with CD44 receptors on neighboring cells. Fibroblasts also produce hyaluronidase, which breaks down hyaluronan into smaller fragments that can be internalized by cells for further use [107]. This balance between the synthesis and degradation of hyaluronan by fibroblasts is essential for maintaining proper tissue homeostasis and facilitating various cellular processes such as wound healing and tissue repair [108].

Patients with poorly controlled type 2 diabetes may have impaired wound healing due to various factors such as peripheral neuropathy, vascular insufficiency, and impaired immune function. Elevated glucose levels in the blood can also damage the small blood vessels and nerves, further impairing the healing process [109]. Hyaluronan (HA) has been shown to promote wound healing in patients with type 2 diabetes [84]. This is because diabetes can impair the body’s natural healing processes, and hyaluronan can help to stimulate and support these processes. Additionally, hyaluronan has anti-inflammatory properties, which can help to reduce inflammation and promote healing in diabetic wounds [110].

The effectiveness of the Healoderm has been evaluated the dressing material containing hyaluronic acid, atelocollagen, and poloxamer, in treating nonischemic-neuropathic DFUs in patients with type 1 or 2 diabetes mellitus who were aged over 20 years [111]. The study participants had an ulcer size of at least 1 cm^2^ that had not shown signs of healing for more than 6 weeks and an ulcer graded as Wagner stage 1 or 2. The participants also had an adequate circulation in the foot, confirmed by the transcutaneous partial pressure of oxygen (TcPO_2_) of at least 30 mm Hg or palpable pulses at the ankle (dorsal pedis artery or posterior tibial artery) and diabetic peripheral neuropathy diagnosed with the Michigan Neuropathy Screening Instrument with a score of at least 2.5. The control group was treated with a conventional moisture-retentive dressing to compare the effectiveness of healoderm. The primary endpoint was the complete healing rate 12 weeks after the commencement of the dressing [112].

### 3.1. Hyaluronan-Based Nanoparticles

Nanomaterials have gained significant attention in biomedical applications due to their unique physicochemical properties, such as high surface area-to-volume ratio, biocompatibility, and tunable surface properties [113]. Different nanomaterials, such as gold nanoparticles (AuNPs) [114], silver nanoparticles (AgNPs) [115], materials-based carbon (graphene, carbon nanotubes (CNTs) [116], and quantum dots (QDs) [117], iron oxide [118], hydroxyapatite [119], zinc oxide [120] and porous composite materials [121], are applied in reactions with hyaluronic acid. Different methods are used to prepare or synthesize hyaluronic acid nanocomposites, such as coating [10], casting film [121], physical, chemical preparation [65], colloidal assembly [122], in-situ preparation [1], enzymatic [123], isotropic gelation [124].

Also, developing such wound dressing materials based on chitosan/hyaluronan composite/nonwoven fabrics is a significant advancement in the field of wound healing, and the results of the study indicate that it has excellent potential as a safe and effective wound dressing material. The lack of cytotoxicity and the acceleration of wound healing properties in animal studies further demonstrate the potential of this novel dressing material [125].

Hyaluronan (HA) based nanocomposites are excellent biomaterials for improving wound healing because they promote tissue regeneration and repair. These nanocomposites can be made by incorporating HA into various materials, such as metals, metal oxides, and polymers, to enhance their properties and effectiveness in wound healing [126]. The HA in these composites can act as a scaffold to support cell growth and proliferation and provide anti-inflammatory and antibacterial properties. Adding metal or metal oxide nanoparticles can also provide antimicrobial and antioxidant activity, aiding healing [127].

This section sheds light on preparing hyaluronan nanocomposite using silver and gold NPs as nanoscale particle models in order to prepare wound dressing. Significant efforts have been devoted generated silver nanoparticles (AgNPs) with excellent distribution with controlled shape and size of the nanoparticles [128]. AgNPs were synthesized without using any external hazard-reducing agents with eco-friendly steps of AgNPs has become a challenging issue for different groups. Nowadays, many research groups have been attracted to integrate green technology in the synthesis of silver nanoparticles (AgNPs) using polysaccharides such as hyaluronan, chitosan, and their derivatives, schizophyllan [129], alginate [130], carboxymethyl cellulose [131], starch [132] and their products, etc. Hyaluronan has been used as a capping/template and reducing agent for the preparation AgNPs. HA/AgNPs composites have been prepared and further developed for the first time as antitumor efficacy [133] due to the unique new physicochemical and optical properties as well as the nontoxicity, spherical/small size, and long-term stability. Also, it can be used as an excellent marker for micro/nano-X-ray computed tomography (Micro/nano CT) [134,135].

Silver nanoparticles (AgNPs) are prepared using sodium hyaluronate as a reducing and stabilizing agent. New hyaluronan/silver nanoparticle composite fibers/fabrics using wet-spinning techniques in two different ways (Figure 4) [15]. First, silver nanoparticles are produced by an in-situ method using different concentrations of silver nitrate as a source of AgNPs. After proceeding with viscous solution from HA/AgNPs nanocomposites, wet spinning techniques are used to prepare fibers/fabrics wound dressing-based HA/AgNPs (Figure 5).

On the other hand, the spongy composites with silver nanoparticles (AgNPs) synthesized by freeze-drying a mixture of silver nitrate (AgNO_3_) and chitosan-l-glutamic acid (CG) derivative loaded with hyaluronic acid (HA) solution have shown promising properties for wound healing. The interconnected porous structure and rough surfaces of the resulting CG/AgNPs spongy composites provide good mechanical properties, swelling, and water retention capacity [136]. In vitro, antibacterial tests have shown that the CG/AgNPs spongy composites effectively inhibit the growth and penetration of bacterial strains such as *Escherichia coli* and *Staphylococcus aureus*. Moreover, the spongy composites containing low concentrations of AgNPs were biocompatible, as they were non-toxic to L929 cells. In vivo tests have also shown that the CG/HA/AgNPs spongy composites promote wound healing, as evidenced by wound contraction ratio, average healing time, and histological examination. These results indicate that the spongy composites with silver nanoparticles have the potential to enhance wound healing by providing both antibacterial and regenerative properties.

Another example of HA-based nanocomposite is hyaluronan/gold (HA/AuNPs) nanocomposite which is synthesized by using gold nanoparticles bearing adamantine moieties and cyclodextrin (CD) modified hyaluronan. The stability of the HA@CD-AuNPs nanocomposite is a manifestation of high interactions between CD cavities and AuNPs. The macromolecular hyaluronan/AuNPs system is subsequently discussed as a drug carrier/delivery for different anticancer drugs, like doxorubicin hydrochloride (DOX), topotecan hydrochloride (TPT) etc. The in-vitro experimental showed that the DOX-impeded HA/AuNPs delivery system exhibited high cellular uptake and anticancer activates comparable to free DOX but with low side effects owing to CD44 receptors mediated endocytosis [137,138].

Other researchers have developed Hyaluronan-based nanocomposites with metal nanoparticles (MNPs) or metal oxide nanoparticles (MO-NPs) have shown promising results in wound healing applications. These nanocomposites have the potential to control infection, promote tissue regeneration, reduce inflammation and oxidative stress, and improve wound closure.

The antimicrobial properties of MNPs or MO-NPs loaded with HA nanocomposites can help to prevent infection at the wound site, which is a major concern in wound healing. These nanocomposites can also promote the growth and proliferation of skin cells, leading to faster wound closure and improved tissue regeneration [139].

Hussain and coworkers [140] published a study in which they developed hyaluronic acid (HA)-functionalized nanoparticles to co-deliver resveratrol and curcumin to treat chronic diabetic wounds. The study aimed to address the limitations of traditional diabetic wound treatments, which often involve multiple therapies and result in poor patient outcomes due to the slow healing process. The researchers hypothesized that combining resveratrol and curcumin, two natural compounds with anti-inflammatory and antioxidant properties, could promote faster healing of chronic diabetic wounds [140].

Fahmy and coworkers [141] published a study in which they formulated a wound dressing made of chitosan-hyaluronic acid (HA) non-woven fabric that was incorporated with silver nanoparticles, and it aimed to address the limitations of traditional wound dressings, which often require frequent changes and can lead to infection and delayed healing. The researchers hypothesized that incorporating silver nanoparticles into the chitosan-HA wound dressing could provide antibacterial activity, enhance wound healing, and reduce the need for frequent dressing changes.

Combining chitosan and HA can provide a scaffold for tissue regeneration and a moist environment conducive to healing [141].

Abdelrahman and coworkers [57] have developed an innovative antibacterial wound dressing with excellent swelling capacity and good mechanical properties, using hyaluronic acid (HA) as both a reducing and stabilizing agent, as well as incorporating zinc oxide nanoparticles (ZnO-NPs) through in situ synthesis (Figure 5). The fact that the ZnO-NPs/HA/PVA composite membrane supports the attachment and growth of normal human dermal fibroblasts and human primary osteogenic sarcoma (Saos-2) without exhibiting toxicity is a significant achievement. It indicates that the composite membrane is biocompatible and suitable for promoting cell proliferation and tissue regeneration (Figure 6).

Moreover, the in vivo measurements of the nanocomposite PVA/HA/ZnO-NP membrane promote infected wound healing compared to the control sample, further highlighting its potential as an antibacterial wound dressing material. This suggests that the incorporated ZnO-NPs effectively contribute to the antibacterial properties of the membrane, helping to prevent or control infections in wounds. The positive results indicate that the nanocomposite membrane has a reasonable potential for application in wound healing. The ability to support cell growth and promote infected wound healing positions this material as a promising option for addressing the challenges associated with wound care [57].

### 3.2. Hyaluronan Injectable Gel

Hyaluronan can also form hydrogels, three-dimensional networks of hydrophilic polymer chains that can absorb and retain large amounts of water [142]. while HA scaffolds can provide a 3D framework for cells to grow and differentiate into functional tissues [143,144]. Furthermore, HA membranes can be used to promote wound healing and tissue regeneration. One of the unique features of hyaluronan is its biodegradability, which is mediated by the action of hydrolases, such as hyaluronidase [57]. Hyaluronidase breaks the glycosidic bond between two residues, allowing for the degradation of hyaluronan. Numerous examples of HA-based hydrogels have been developed for use in medical fields. A novel series of N-succinyl chitosan-oxidized hyaluronic acid-based (NSC-OHA-based) hydrogels were designed for use as injectable self-healing hemostatic agents and wound dressings. These hydrogels were developed by incorporating calcium ions (Ca^2+^) and/or a four-armed amine-terminated poly(ethylene glycol) (4-arm-PEG-NH_2_ or PEG1) into the NSC-OHA hydrogel formulation. The study found that the modified hydrogels exhibited improved biocompatibility, hemostasis, and wound healing properties compared to the NSC-OHA hydrogel alone, with the calcium-containing hydrogels showing the most promising results [46]. The hydrogels could be formed within 30–60 s at room temperature and could be injected smoothly through a 23G needle, which has an outer diameter of 0.6 mm. This injectability and ease of use make the hydrogels suitable for clinical applications.

Hydrogels based on collagen I and hyaluronic acid (HA) have been used to encapsulate and deliver neural stem cells for the treatment of spinal cord injury and to support the growth and differentiation of neurons in vitro [145,146,147].

HA hydrogels have also been used to deliver drugs and other therapeutic agents to the brain, provide mechanical support, and promote tissue regeneration following traumatic brain injury [148]. One of the main advantages of using HA as hydrogel is its dual functionality, which allows it to be modified with reactive groups such as carboxyl and hydroxyl groups. This makes grafting functional groups onto the HA molecule possible, which can then be used to form chemical or physical hydrogels through cross-linking reactions between the reactive groups and cross-linkers such as EDC, GLU, or ECH. HA in hydrogels represents a promising approach for developing new therapies for neurological disorders and injuries and studying neural development and function in vitro. HA-based hydrogels have also been studied for use in localized cancer treatment. Due to their biocompatibility and targeting properties, they can be loaded with anti-cancer drugs or other therapeutic agents and delivered directly to the tumor site, minimizing systemic toxicity and improving treatment efficacy [149]. A hydrogel based on collagen I and hyaluronic acid (HA) was fabricated in a study, and the two components were covalently cross-linked to each other under mild HRP (horseradish peroxidase) catalysis [135,138,139]. The resulting hydrogel had a porous structure that contributed to its ability to retain water, exchange gases, allow for nutrition penetration, and facilitate cell dwelling. This makes hydrogel potentially useful for applications such as tissue engineering, wound healing, and drug delivery. In addition, The COL-HA hydrogel fabricated in a study allowed vascular cells to grow, indicating its good biocompatibility and biodegradability. When used to treat wounds, the hydrogel promotes the formation of vasculature, epithelial layers, and collagen fibers, leading to quicker healing. The study demonstrated that the COL-HA hydrogel was an effective wound-healing platform without the need for additional bioactivities. Its ability to promote cell growth and tissue regeneration, as well as its biocompatibility and biodegradability, make it a promising material for wound healing applications. The formation of the vasculature, epithelial layers, and collagen fibers is essential for proper wound healing, and the COL-HA hydrogel was found to promote all three. This suggests that the hydrogel may improve healing outcomes for various wounds, such as chronic wounds or wounds in patients with compromised healing abilities.

Another study developed hydrogels by combining alginate (ALG) and hyaluronan (HA) as a biofunctional platform for dermal wound repair [150]. The hydrogels were produced using internal gelation and were found to be homogeneous and easy to handle. Alginate and hyaluronan are natural polysaccharides with unique properties that make them suitable for biomedical applications [150]. Alginate is biocompatible and forms gels in the presence of divalent cations, while hyaluronan is involved in cell migration, proliferation, and wound healing. By combining these two polysaccharides in a hydrogel, the resulting material had the potential to mimic the natural extracellular matrix and promote cell growth and tissue regeneration. The hydrogels produced by internal gelation were homogeneous and easy to handle, which is important for their use in clinical settings. The study found that the ALG-HA hydrogels had good mechanical properties, swelling behavior, and biodegradability, making them suitable for use in dermal wound repair. The hydrogels also supported cell attachment and proliferation, indicating their biocompatibility [150].

Also, studies have shown that combining hyaluronic acid and poloxamer 407 in a hydrogel has several advantages for wound healing. The hydrogel can transform from a liquid to a gel at body temperature, conforming to the shape of the wound and providing a moist environment that is conducive to healing. The hydrogel also has excellent moisturizing properties, which can help prevent the wound from drying out and promote cell growth. Additionally, the hydrogel can increase protein accumulation in the wound area and allow for greater air permeability than a typical wound covering. It also has been found to be effective in isolating skin wounds from bacterial invasion, which can help reduce the risk of infection and promote faster healing [151,152].

Another example of injectable hydrogel is fabricated from hyaluronan with low molecular weight grafted by polyethylene glycol derivatives via the Shiff base mechanism [77]. The injectable soft hydrogels show good elastic characteristics and self-healing properties (Figure 7). The injectable soft gel based on marine hyaluronan is a promising biomaterial and can be employed as a new wound healing hydrogel with high-performance properties (Figure 7).

Another injectable hydrogel is based on marine hyaluronan derivatives grafted with thiolated poly(γ-glutamic acid) [153]. Grafting the oxidized hyaluronic acid with poly(γ-glutamic acid) improves the prepared hydrogel’s mechanical properties, enhancing the fibroblasts cell growth and accelerating the wound healing ability (Figure 8).

### 3.3. Hyaluronan 3D Scaffolds

Hyaluronan-based scaffolds can be produced by a variety of methods, including electrospinning [154], phase separation, bioprinting [155], supercritical fluid technology, porogen leaching, centrifugal casting, freeze-drying, micro-pattering, and cross-linking techniques. These methods allow for the creation of scaffolds with varying pore sizes, shapes, and mechanical properties, which can be tailored to specific applications.

Electrospinning is a commonly used method for producing hyaluronan-based scaffolds with high porosity and specific surface area. In this technique, a polymer solution is subjected to an electric field, and a fine jet of polymer is spun and deposited onto a collector, forming a fibrous scaffold. The scaffold’s pore size and mechanical properties can be controlled by adjusting the spinning conditions, such as polymer concentration, voltage, and distance between the spinneret and collector. Freeze-drying is another method for producing hyaluronan-based scaffolds. In this technique, a hyaluronan solution is frozen and then lyophilized to remove the water, leaving behind a porous scaffold. The scaffold’s pore size and mechanical properties can be controlled by adjusting the freezing conditions, such as the rate of cooling and the temperature [156]. Crosslinking is a method for stabilizing hyaluronan-based scaffolds and improving their mechanical properties [157]. Crosslinking can be achieved using chemical crosslinkers or physical crosslinking, such as UV or heat treatment. The degree of crosslinking can be controlled by adjusting the concentration of crosslinkers or the duration of treatment.

Hyaluronan-based scaffolds have shown promising results in various preclinical and clinical studies [158]. The biocompatibility and biodegradability of hyaluronan make it an attractive material for tissue engineering applications. As Hyaluronan is a natural component of the extracellular matrix (ECM) in many tissues, including skin, cartilage, and connective tissue, hyaluronan-based scaffolds can mimic the native ECM, providing a suitable environment for cells to proliferate, differentiate, and organize [158]. In wound healing, it can be used to support the growth of new tissue and promote wound closure, and in cartilage repair, it can be used to support the development and organization of chondrocytes, the cells that produce cartilage.

One of the most promising applications of HA-based scaffolds is bone tissue engineering. Bone defects and fractures are common clinical problems, and current treatment options, such as autografts and allografts, have limitations [159]. HA-based scaffolds have been shown to support osteoblasts’ adhesion, proliferation, and differentiation, the cells responsible for bone formation [160]. They also can release growth factors such as bone morphogenetic protein (BMP), which can further enhance bone regeneration. In cartilage tissue engineering, HA-based scaffolds have been used to support the growth of chondrocytes, the cells responsible for producing cartilage tissue. These scaffolds can be modified to provide mechanical properties similar to that of native cartilage and can also release growth factors such as transforming growth factor beta (TGF-β) to enhance cartilage regeneration [161].

HA can also be combined with other biomaterials, such as collagen and chitosan, to enhance their mechanical properties and improve tissue integration. HA has been utilized as a drug delivery system for targeted therapy in cancer treatments. It has been shown to accumulate in tumor tissues due to the increased expression of HA receptors on cancer cells. By conjugating chemotherapeutic agents to HA, it is possible to selectively target cancer cells and reduce the systemic toxicity of the drug [69]. The extracellular matrix (ECM) is a complex network of proteins and other molecules surrounding cells in tissues and organs, providing structural support, and regulating various cellular processes. In the nervous system, the ECM actively contributes to growth and maturation and has a wide range of regulatory functions, as pointed out by several authors [162]. Hyaluronic acid (HA) is a key component of the ECM and has been extensively studied for its potential applications in the nervous system. HA is a large, negatively charged polysaccharide that is widely used in the preparation of hydrogels due to its unique properties [142].

Scaffold-based marine hyaluronan and silk fibroin (SF) is prepared via electrospinning. Different ratios between HA and SF are blended in an aqueous solution medium (Figure 9). The effect of hyaluronan molecular weight on scaffold morphological properties shows promising results, and the biocompatibility improved in the presence of marine hyaluronan. The prepared materials are promising for tissue engineering, especially skin healing applications [163].

### 3.4. Hyaluronan Membranes

Hyaluronan-based membranes can be produced using various techniques, such as electrospinning, solution casting, or phase separation. These membranes can be composed solely of HA or combined with other polymers or materials to enhance their properties. The specific composition and structure of the membranes can be tailored to meet the requirements of different applications. These membranes have been extensively studied and applied in various biomedical fields, including tissue engineering, drug delivery, wound healing, and regenerative medicine. They offer several advantages, such as promoting cell adhesion, proliferation, and migration and providing a favorable microenvironment for tissue regeneration [164,165,166].

The researchers explored the potential of using bacterial cellulose (BC) and hyaluronan (HA) to develop BC-HA nanocomposite films for wound dressing applications. They found that the BC-HA composite films exhibited improved properties compared to pure BC, including better water uptake capability, and meeting the requirements for breathability. In terms of mechanical properties, the BC-HA composite films had higher elongation at the breakpoint, indicating increased flexibility, but their tensile strength decreased with higher HA content. This suggests that an optimal concentration of HA is needed to balance the mechanical properties of the composite films. Furthermore, the BC-HA composite films showed low toxicity and promoted the growth of primary human fibroblast cells, indicating their biocompatibility. In vivo experiments on animals demonstrated that the BC-HA composite films with 0.1% HA had the shortest wound healing time, while those with 0.05% HA resulted in the best tissue repair outcomes. Based on these findings, the researchers concluded that BC-HA composite films have the potential to serve as effective wound dressing materials for clinical skin repair. The combination of BC’s mechanical properties and HA’s healing properties make these nanocomposites promising for promoting wound healing, reducing scarring, and creating an optimal environment for tissue regeneration [167].

Karine and coworkers [168] utilized the electrospinning technique to create cross-linked hyaluronic acid (HA)/poly (vinyl alcohol) (PVA) membranes. The primary challenge in electrospinning HA is its rheological characteristics. To address this, the researchers combined HA with PVA, which has favorable rheological properties for electrospinning. Their objective was to develop HA/PVA membranes without using organic solvents and achieve cross-linking through a photo crosslinking process. The goal was to produce biocompatible membranes suitable for drug delivery applications. The results of the study showed successful electrospinning under all tested conditions. However, complete cross-linking was observed only when using 15% and 30% crosslinker concentrations, which was confirmed by infrared spectroscopy. The inclusion of the crosslinker improved the stability of the electrospinning process, particularly at a 30% concentration. The membranes demonstrated no cytotoxicity even after cross-linking, indicating their potential as drug-delivery devices. The absence of organic solvents during fabrication, combined with the low degradation rate of the membranes in PBS at pH 7.4, further supported their suitability for biomedical applications [168].

New nanofibers with a layered composition of polylactide (PLA) and sodium hyaluronate (HA) using the electrospinning method. They hypothesized that these nanofibers would serve as a suitable matrix for cell adhesion and proliferation. The study aimed to determine the optimal electrospinning conditions for creating the layered compositions and assess the nanofibers’ biocompatibility with fibroblasts. To establish the optimal electrospinning conditions, various parameters such as solvent amount, polymer concentration, mixing temperature, and electrospinning process conditions were modified. By adjusting these parameters, the researchers were able to control the diameter and properties of both the PLA and HA fibers. The spinning solution was characterized using surface tension and rheology measurements. The morphology and fiber diameters of PLA and HA were examined using a scanning electron microscope (SEM). The PLA/HA nonwoven structure was analyzed using spectroscopy (FTIR/ATR). The biocompatibility of the PLA/HA nonwoven with fibroblasts, which are extracellular matrix (ECM) producers, was assessed under in vitro conditions. The results demonstrated stable conditions for forming submicron PLA fibers using a 13% wt. solution of the polymer dissolved in a 3:1 mixture of dichloromethane (DCM) and dimethylformamide (DMF) at 45 °C. Hyaluronic acid fibers were prepared from a 12% wt. solution of the polymer dissolved in (2:1) mixture of ammonia water and ethyl alcohol. All materials exhibited biocompatibility, although to varying degrees [169].

In conclusion, the proposed laminate scaffold comprised a hydrophobic-hydrophilic domain surface with maintained fiber size in both layers. The material showed positive results in terms of biocompatibility when tested with fibroblasts. These findings suggest that the PLA/HA nanofiber scaffold has potential as a matrix for cell adhesion and proliferation, indicating its suitability for applications in tissue engineering and regenerative medicine [170].

## 4. Conclusions

The field of wound management has recognized the importance of providing external biological and physiological nourishment to support the wound-healing process. Current research has demonstrated the significant potential of marine-derived biological macromolecules, such as hyaluronan, in enhancing wound healing and promoting the regeneration of skin tissue. Here are some key points regarding the use of marine hyaluronan macromolecules, and it plays a crucial role in various biological processes, including wound healing. Hyaluronan-based materials have been studied extensively for their ability to provide a favorable environment for wound healing. They can help maintain optimal moisture levels, promote cell migration and proliferation, and modulate inflammatory responses.

Marine hyaluronan macromolecules have been incorporated into skin tissue-engineered substitutes fabricated using different synthetic polymers or bioactive molecules. These substitutes can serve as scaffolds to support cell growth and tissue regeneration. The inclusion of marine hyaluronan further enhances the efficiency and compatibility of these materials. Research has confirmed that marine-derived hyaluronan macromolecules are cytocompatibility, meaning they are well-tolerated by cells and tissues. They have also been shown to promote wound healing and skin tissue regeneration efficiently. These properties make marine hyaluronan valuable in wound dressings, scaffolds, and other wound care products. By leveraging the benefits of marine hyaluronan and other marine-derived biological macromolecules, researchers and clinicians can continue to advance wound management techniques and develop innovative solutions to improve wound healing outcomes and facilitate the reestablishment of healthy skin tissue.

## 5. Future Trend and Beyond Limitations

The ultimate goal of marine-derived tissue-engineered skin scaffolds is to replicate the functions of native tissue. Addressing key tissue parameters during scaffold engineering is crucial as they serve as templates for new tissue formation and provide guidance for cell growth. As marine extracts continue to be integrated with various synthetic or natural materials for designing skin tissue substitutes, comprehensive studies are needed to overcome limitations associated with the scaffold’s mechanical and biological properties. Additionally, it is important to investigate the limitations of scaffold fabrication techniques, particularly regarding the use of solvents and cross-linking chemicals.

The use of solvents and cross-linking chemicals in scaffold fabrication can introduce toxic residuals that may have detrimental effects on cellular activities. Therefore, alternative materials and methods should be explored to fabricate novel scaffolds with marine molecules, addressing these concerns and ensuring the safety and efficacy of the final product. The use of hydrogel, membrane and scaffolds based on marine biomaterials as a delivery mode for bioactive molecules is indeed gaining significant attention, as it allows for targeted delivery and interaction with the site of importance. In this context, it is proposed to explore and isolate biologically active molecules from marine sources and incorporate them into different biomaterials shapes. By containing isolated bioactive compounds in marine macromolecule-derived tissue-engineered skin substitutes, it is possible to develop effective approaches for wound healing treatment. Modifying these substitutes with bioactive compounds from marine sources offers potential benefits in promoting wound healing.

To fully understand the mechanism of action of the isolated compounds on wound healing, it is essential to conduct critical investigations. These investigations should focus on evaluating the effects of the compounds when combined with different marine biomaterials. By studying the interactions between the bioactive compounds and the marine matrix, researchers can gain insights into how they influence wound healing. Both in vitro and in vivo investigations are crucial for comprehensively assessing the biological acceptance and efficacy of the modified marine matrix. In vitro, studies allow for controlled experiments to examine cellular responses and interactions, while in vivo studies provide a more realistic evaluation of the marine-based matrix performance in an animal or human model. Conducting further in vitro and comprehensive in vivo investigations will offer a better understanding of the biological acceptance of the marine-based matrix and the efficacy of the incorporated bioactive compounds. This knowledge is essential for advancing wound healing research and developing innovative approaches for effective treatments.

In conclusion, the investigation and isolation of biologically active macromolecules like hyaluronan, chitosan, collagen, chitin, and alginate from different marine sources, their incorporation into different supporting materials (natural or synthetic), and the comprehensive study of their effects on wound healing, combined with in vitro and in vivo investigations, hold great potential for advancing the field of tissue engineering and improving wound healing outcomes.

## Figures and Tables

**Figure 1 marinedrugs-21-00426-f001:**
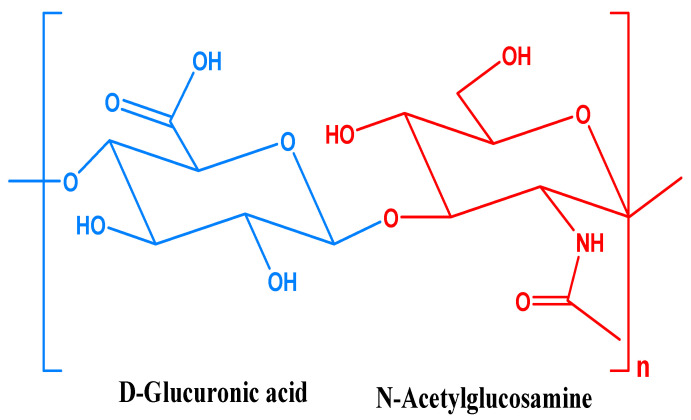
Chemical structure of hyaluronic acid.

**Figure 2 marinedrugs-21-00426-f002:**
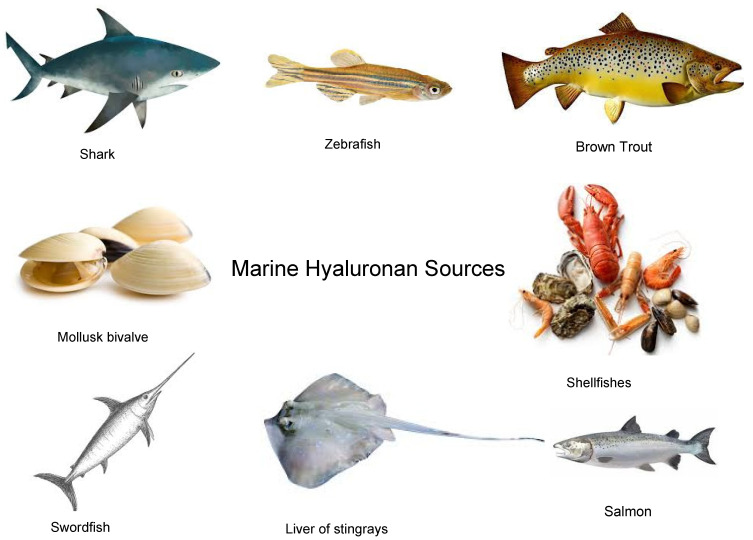
Different marine sources of hyaluronic acid.

**Figure 3 marinedrugs-21-00426-f003:**
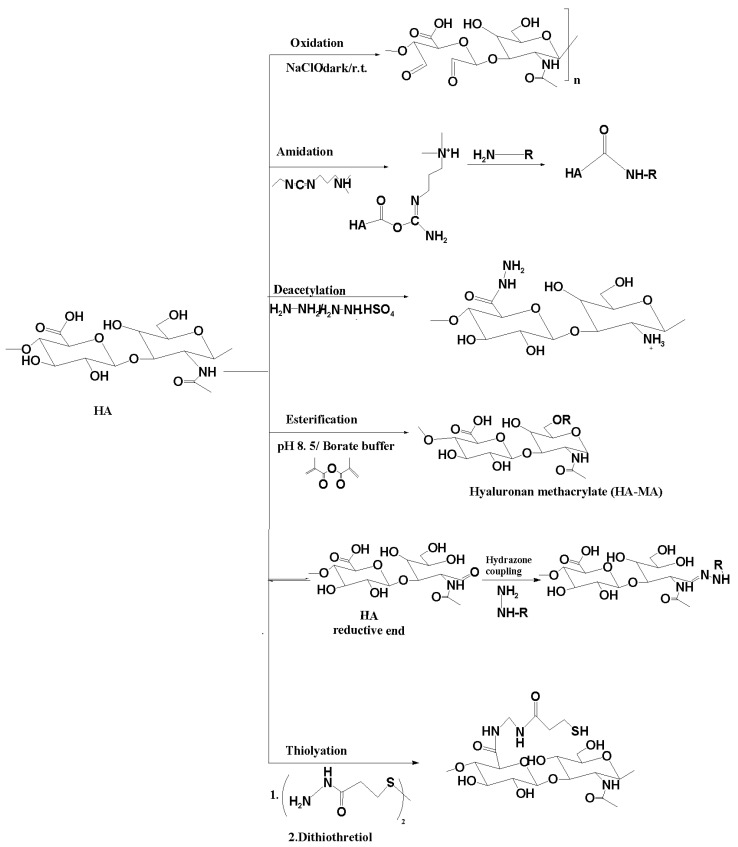
Chemical modifications of hyaluronan.

**Figure 4 marinedrugs-21-00426-f004:**

Wound Healing Phases.

**Figure 5 marinedrugs-21-00426-f005:**
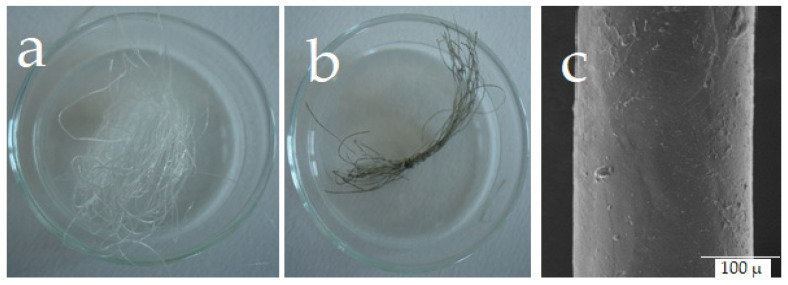
Preparation of hyaluronan fibers with silver nanoparticles. Photographs of native hyaluronan fibers (**a**), HA fibers with Ag^0^ (**b**), SEM of Ag^0^ @HA (**c**).

**Figure 6 marinedrugs-21-00426-f006:**
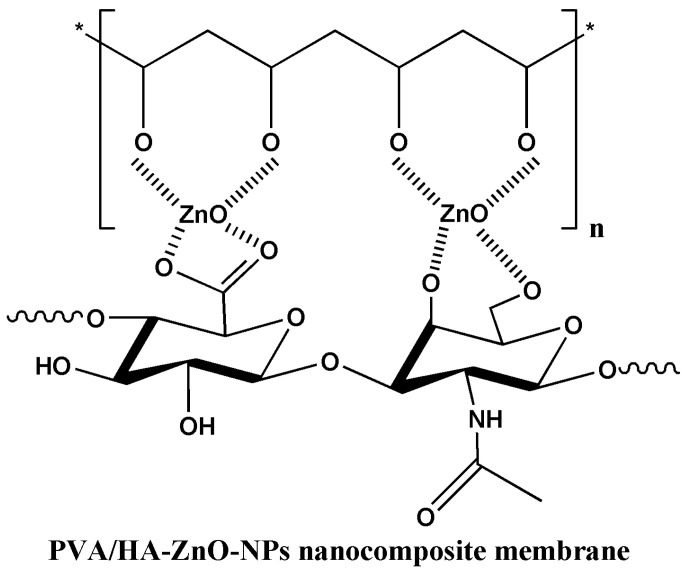
In-situ synthesis of zinc oxide nanoparticles using hyaluronan and polyvinyl alcohol.

**Figure 7 marinedrugs-21-00426-f007:**
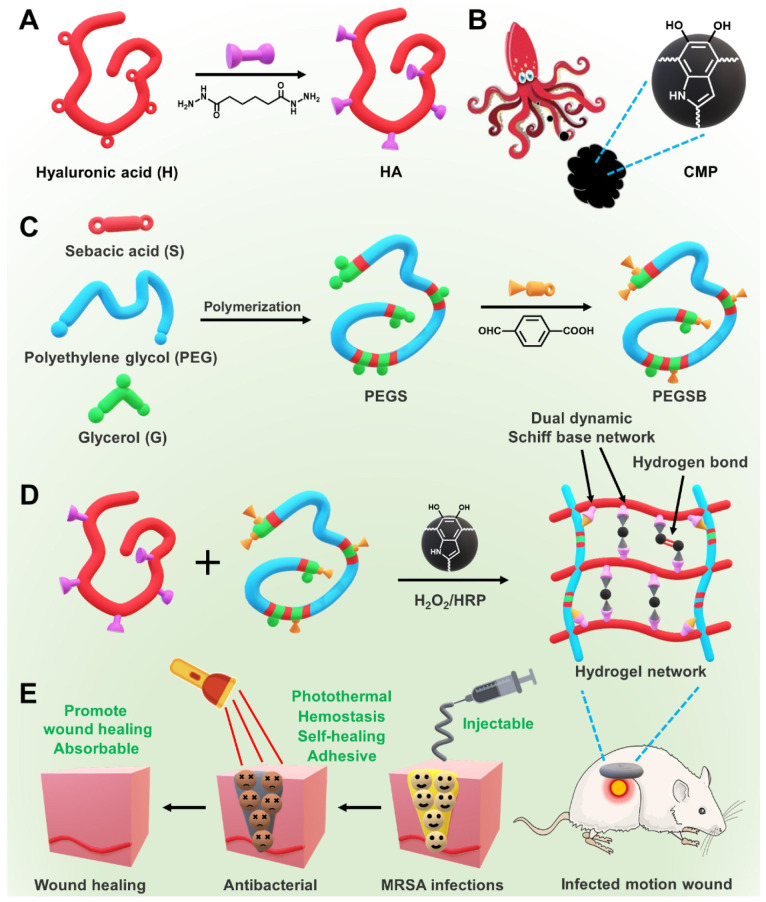
Schematic diagram of HA-PEGSB-CMP hydrogel fabrication. (**A**): The synthesis of HA derivatives. (**B**): Chemical structure of CMP. (**C**): The synthetic way of PEGSB. (**D**): representation of HA-PEGSB-CMP hydrogel synthesis with dual crosslinking ways. (**E**): The application diagram of the hydrogels treating infected motion wound and promoting wound healing Reproduced with permission from ref. [77].

**Figure 8 marinedrugs-21-00426-f008:**
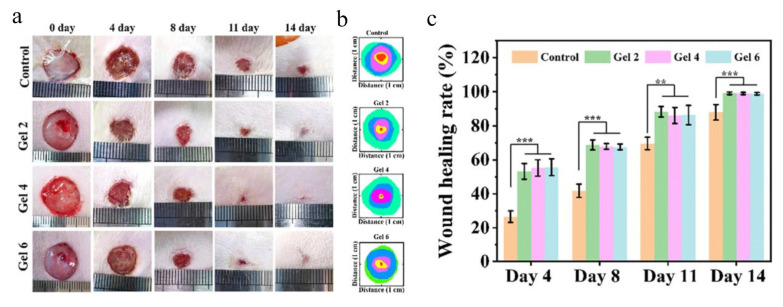
Macroscopic evaluation of wound healing in the defected skin of rats. Photographs of wounds at different day treatment (**a**), Description of the wound healing process (**b**); Wound closure percentage (**c**) (** *p* < 0.01, *** *p* < 0.001). Reproduced with permission from ref. [153].

**Figure 9 marinedrugs-21-00426-f009:**
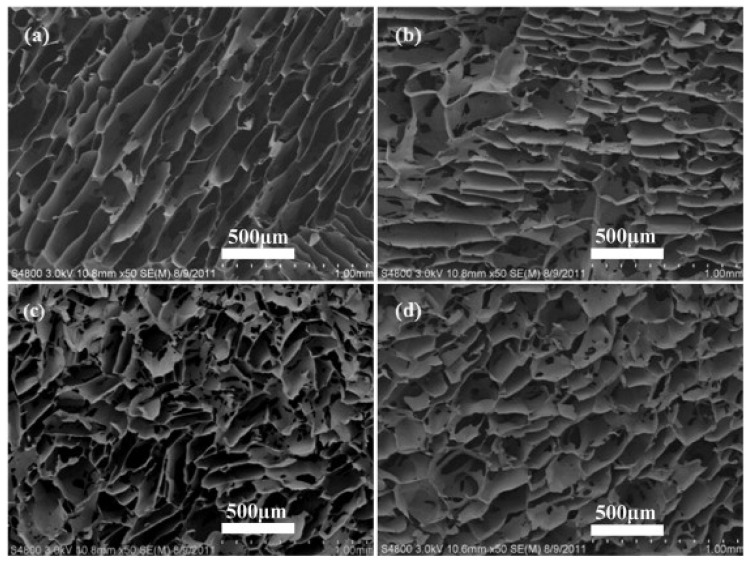
Microstructure of the silk fibroin/hyaluronan 3D scaffolds. Different MW HA: (**a**) pure SF; (**b**) HA 0.6 × 10^6^ Da; (**c**) HA 1.6 × 10^6^ Da; (**d**) HA 2.6 × 10^6^ Da. Reproduced with permission from ref. [163].

**Table 1 marinedrugs-21-00426-t001:** The yield percentage of HA depends on the source.

Terrestrial Sources		Marine Sources		Ref.
Synovial fluid	250 mg/L	Swordfish	55 mg/L	[31]
Bovine	0.47 mg/L	Shark	300 mg/L	[48]
Pig	40 mg/L	Tuna	420 mg/L	[49]

**Table 2 marinedrugs-21-00426-t002:** Marine hyaluronan-based wound dressings for several types of wound repair.

HA MWt	Another Additive	Shape Form	Preparation Method	Properties	Medical Purposes	Ref.
LMW-HA	Pollulan	Film	grafting	Anti-enzymatic degradation	Treatments of skin defect	[75]
LMW-HA	Peptides	Hydrogel	grafting	Antibacterial and Injectable	Wound healing	[76]
MMW-HA	PEGSB	Hydrogel	grafting	Enhance tissue adhesion	Treatments of skin defect	[77]
LMW-HA	ADP	Scaffold	blend	Haemostatics performance	Wound healing	[78]
MMW-HA	CO	Hydrogel	grafting	High repair effect	Burn wound treatment	[79]
HMW-HA	ADM	Hydrogel	grafting	Anti-scarring activity	Treatments of burn skin	[80]
MMW-HA	CMC	Hydrogel	grafting	Anti-inflammation	Diabetic wound treatment	[81]
LMW-HA	Fe^3+^	Hydrogel	ionic	Anti-inflammation	Diabetic wound treatment	[82]
HMW-HA	DFO	Hydrogel	enzymatic	Promoted angiogenesis	Diabetic wound treatment	[83]
HMW-HA	Chitosan	Sheet	grafting	Increased collagen deposition	Diabetic wound treatment	[84]
MMW-HA	Chitosan	Hydrogel	grafting	pH-response	Diabetic wound treatment	[85]
HMW-HA	Chitosan	Mat	blend	Grafting	Wound healing	[86]
MMW-HA	Pectin	hydrogel	ionic	Self-healing	Wound treatment	[87]
LMW-HA	CMS	hydrogel	chemically	Antibacterial, Injectable	diabetic wound healing	[88]
MMW-HA	EPL	Hydrogel	chemically	wet adhesion, self-healing	Wound healing	[89]

## Data Availability

Data are contained within the article.

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
