# Peer review of "Marine Biomaterials: Hyaluronan"

_marinedrugs, 2023, doi:10.3390/md21080426_

Round 1

Reviewer 1 Report

In the manuscript, R. M. Abdel-Rahmana and A. M. Abdel-Mohsena have intended to comprehensively review the hyaluronic acid, a versatile polymer originated from marine sources, in wide range of aspects, from extraction sources, structure, modification, and biomedical applications. The work is quite useful and interesting. However, it is hard to consider for publication, the manuscript exists a lot of issues that need to seriously addressed. I suggest some of revision as follow:

1. I recommend the authors should re-organize the structure of the manuscript so that the reader can easily follow. If the authors want to focus on application of hyaluronan for wound healing/dressing purposes, please writing separate parts base on physical forms of products for wound healing/dressing, such as injectable system, hydrogel, membrane, electrospinning sponge, scaffold. In my opinion, the author should remove the ‘hyaluronan nanocomposite’ and move this content in proper parts of injectable system, hydrogel, membrane, sponge, scaffold. Simultaneously, the work should skip the studies relating to cancer treatment. More importantly, the number of studies relating to wound healing/dressing referenced in the manuscript too modest and almost them are not really attractive.

2. In the abstract, the first sentence and second sentence are not link well, it must be rewrite.

3. In the introduction, why the number list is only 1.1, the authors should expand the content of this part. Besides, the authors also need to clarify particular GAGs (hyaluronic acid, heparin and chondroitin sulfate) and their characteristics. The advantages of marine originated GAGs should be mentioned.

4. The Fig.1 showed structure of hyaluronic acid which is not related to the sentence used to describe for Fig.1 ‘ Currently, hyaluronan production is performed on a large scale using different methods and sources (Fig. 1)’

5. The authors need to compare in detail the advantages of every method for isolating the hyaluronic acid. The table 1 is too lack of information to understand, for example which part of tuna for extract hyaluronic acid.

6. In the 2.3 part, no information of molecular weight of marine originated hyaluronic acid.

7. In the 2.4 part, the interaction between hyaluronic acid and cells need to provided.

8. The 2.6 part should be move to 'Application part' for better visualization.

9. Please capitalized letter ‘d’ in ‘D-glucuronic acid.

Author Response

Responses to reviewers' comments

In the manuscript, R. M. Abdel-Rahman and A. M. Abdel-Mohsen have intended to comprehensively review the hyaluronic acid, a versatile polymer originated from marine sources, in wide range of aspects, from extraction sources, structure, modification, and biomedical applications. The work is quite useful and interesting. However, it is hard to consider for publication, the manuscript exists a lot of issues that need to seriously addressed. I suggest some of revision as follow:

  1. I recommend the authors should reorganize the structure of the manuscript so that the reader can easily follow. If the authors want to focus on application of Hyaluronan for wound healing/dressing purposes, please writing separate parts base on physical forms of products for wound healing/dressing, such as injectable system, hydrogel, membrane, electrospinning sponge, scaffold. In my opinion, the author should remove the 'hyaluronan nanocomposite' and move this content in proper parts of injectable system, hydrogel, membrane, sponge, scaffold. Simultaneously, the work should skip the studies relating to cancer treatment. More importantly, the number of studies relating to wound healing/dressing referenced in the manuscript too modest and almost them are not really attractive.

Responses:

Thank you for your valuable comments. The authors accepted the reviewer suggestion and reorganized the manuscript accordingly, adding new references focusing on wound healing/dressing based on Hyaluronan.

  1. In the abstract, the first sentence and second sentence are not link well, it must be rewrite.

Response: The first two sentences were corrected and linked according to the reviewer suggestion.

  1. In the introduction, why the number list is only 1.1, the authors should expand the content of this part. Besides, the authors also need to clarify particular GAGs (hyaluronic acid, heparin and chondroitin sulfate) and their characteristics. The advantages of marine originated GAGs should be mentioned.

Response: authors corrected according to reviewers' suggest

  1. The Fig.1 showed structure of hyaluronic acid which is not related to the sentence used to describe for Fig.1 'Currently, hyaluronan production is performed on a large scale using different methods and sources (Fig. 1)'

Response: the sentence was corrected in the text

  1. The authors need to compare in detail the advantages of every method for isolating the hyaluronic acid. The table 1 is too lack of information to understand, for example which part of tuna for extract hyaluronic acid.

Response: The authors already add more info regarding the advantage of the isolation process

  1. In the 2.3 part, no information of molecular weight of marine originated hyaluronic acid.

Response: The authors added hyaluronan molecular weight

  1. In the 2.4 part, the interaction between hyaluronic acid and cells need to provided.

Response: the interaction between Hyaluronan and cells is more described in the text.

  1. The 2.6 part should be move to 'Application part' for better visualization.

Response: Part 2.6 already moved to forms and applications of marine Hyaluronan

  1. Please capitalized letter 'd' in 'D-glucuronic acid.

Response: already corrected in the text

Reviewer 2 Report

This review article comprehensively discussed marine hyaluronan in regard to its sources, production, chemical structures, chemical modifications, biological properties, and biomedical applications. The authors did a good job in putting together the last 20 years of research on hyaluronan and its applications. I enjoyed reading the manuscript. This manuscript will be certainly suitable for publication after some minor revisions.

1.     The figures need more clarity and better quality.

2.     In the introduction part author should include the business aspect of hyaluronan in brief. The approximate market size and industry-wise use of hyaluronan. This information will give the readers the essence of how valuable is hyaluronan as a biomaterial.

Author Response

Comments and Suggestions for Authors

This review article comprehensively discussed marine Hyaluronan in regard to its sources, production, chemical structures, chemical modifications, biological properties, and biomedical applications. The authors did a good job in putting together the last 20 years of research on Hyaluronan and its applications. I enjoyed reading the manuscript. This manuscript will be certainly suitable for publication after some minor revisions.

  1. The figures need more clarity and better quality.

 Response: Thank you. The quality of the figures already improved.

  1. In the introduction part author should include the business aspect of Hyaluronan in brief. The approximate market size and industry-wise use of Hyaluronan. This information will give the readers the essence of how valuable is Hyaluronan as a biomaterial.

Response: Thank you. One paragraph has already been added to the text in the introduction part.

Reviewer 3 Report

Title: Review Marine Biomaterials: Hyaluronan

Comment:

The authors present a review on Hyaluronan, a topic on which they demonstrate good expertise. However, papers are often cited as if they specifically dealt with the characteristics of biomaterials of marine origin while they do not mention them or mention them briefly, without underlining their importance and without proposing a comparison with biomaterials of different origins. Just to give just a few examples limited to the first few lines of the Introduction:

Line 23 The vast diversity of organisms living in the OCEANS presents a wealth of potential for developing high-value bioactive substances and biomaterials [1-3].

Line 24 Although many biomaterials have been derived from MARINE organisms in recent decades [4, 5],

Line 26 MARINE biomaterials have unique properties that make them promising materials for various biological and biomedical applications [6, 7]

Line 28 common MARINE polysaccharides and proteins, are important considerations in the development of these materials [8, 9].

Quickly reading the works cited, the papers [1-6, 8] do not seem to deal with either marine organisms or biomaterials of marine origin, except for occasional mentions of alginates.

Continuing this list is unnecessary, but I would suggest citing more relevant articles that underline the peculiarity of biopolymers of marine origin. I report a few below, but it will not be difficult for the authors to find others that are equally or more valid:

doi.org/10.3390/md18120589

doi.org/10.3390/md20060372

doi.org/10.3390/md20040219

doi.org/10.1021/acs.biomac.1c00013

doi.org/10.1016/j.isci.2023.106404

doi.org/10.1016/j.jece.2021.105895

doi.org/10.1016/j.marenvres.2016.03.007

doi.org/10.3390/polym13152482

Line 3 Use superscript for a*

Line 28 “are important considerations in the”

Do the authors mean important details (or aspects)?

Line 37 I would not use a dedicated subparagraph for 1.1. Glycosaminoglycans

Line 109 Figure 2 needs to be redesigned because only bivalve molluscs show the correct magnification

Lines 212-214 Replace “For example, HA hydrogels can be used as injectable matrices to deliver cells or drugs directly to a specific tissue site[58].. For example, HA injections are commonly used in cosmetic procedures to” with “For example, HA hydrogels can be used as injectable matrices to deliver cells or drugs directly to a specific tissue site [58] and HA injections are commonly used in cosmetic procedures to” for avoid useless repetitions.

Lines 294-297 The marine hyaluronan-based wound healing used in three typical wound repair types [74]. The marine hyaluronan-based wound healing used in three typical wound repair types is listed below in Table 2.

Comment: Either the sentence was repeated unnecessarily or something is missing in the first sentence. Also in this case, the cited paper does not mention material of marine origin except as an incidental mention of alginates but without clarifying how they differ from other biomaterials of different origins.

Line 307 Enlarge the firt box of Figure 4 to not separate the letter n from the word inflammatio

Line 319 Replace “tissue repair. the inflammation” with “tissue repair. The inflammation”

Line 371 Use superscript for 2 in cm2

Line 374 Use subscript for 2 in (TcPO2)

Line 450 Use italics for in vitro

Line 463 Replace “Hussain et al [140]. published a study” with “Hussain and coworkers [140] published a study”

Line 471 Replace “Fahmy et al [141]. published a study” with “Fahmy and coworkers [141] published a study”

Line 480 Replace “Abdelrahman et al [142] have developed“ with “Abdelrahman and coworkers [142] have developed”

Line 511 Use superscript for 2+ in (Ca2+)

Line 515 Replace the comma with a point in “most promising results [46], The”

Line 520 “Another type of HA-hydrogel is in the nervous system”.

Comment: I don't understand what the authors mean by this sentence. Something is missing?

Line 522 Use italics for in vitro.

Line 522 Replace “[146, 147] [148].” with “[146-148].”

Line 530 Replace “the use of HA“ with “The use of HA”

Line 532 Use italics for in vitro

Line 536 A hydrogel based on collagen I and hyaluronic acid (HA) was fabricated in a study,

Comment: I think it is appropriate to insert a citation here.

Line 553 “In another study, hydrogels were developed by combining alginate (ALG) and hya-“

Comment: If the relevant citation is [151], it must be inserted immediately, if it has been omitted, add it.

Line 567 Replace “Also, Studies have shown” with “Also, studies have shown”

Line 588 “Another injectable hydrogel based on marine hyaluronan derivatives” with “Another injectable hydrogel is based on marine hyaluronan derivatives”

Lines 652-655 “The effect of hyaluronan molecular weight on scaffold morphological properties shows promising results and the biocompatibility improved in presence of marine hyaluronan. The prepared materials are promising for tissue engineering application especially for skin healing applications[164].”

Comment: It does not appear that the hyaluron mentioned in this paper, [164], was of marine origin.

Line 680 Use italics for in vivo

Line 688 Replace “In Karine et al [169] utilized the electrospinning technique” with “Karine and coworkers [169] utilized the electrospinning technique”

Line 716 Use italics for in vitro

Lines 775-777 “These investigations should focus on evaluating the effects of the compounds when combined with different marine as a matrix.”

Comment: The sentence seems incomplete. Did the authors intend to write marine biopolymers, marine biomaterials or something like that?

Line 779 Use italics for in vitro and in vivo

Line 780 Use italics for in vitro

Line 781 Use italics for in vivo

Line 783 Use italics for in vitro and in vivo

Line 752 The Future Trend and Beyond Limitations paragraph appears a bit repetitive. I think it could be summarized without much difficulty.

Lines 797-799 Abdel-Mohsen, A. M.; Jancar, J.; Abdel-Rahman, R. M.; Vojtek, L.; Hyršl, P.; Dušková, M.; Nejezchlebová, H., A novel in situ silver/hyaluronan bio-nanocomposite fabrics for wound and chronic ulcer dressing: In vitro and in vivo evaluations. International Journal of Pharmaceutics 2017, 520, (1), 241-253.

Comment: The first citation must be formatted in accordance with the rest of the bibliography. Also italics should be used for in situ, in vitro and in vivo.

Line 869-870 Use italics for (Lapemis curtus):

Line 891 Do not use capital letter for ISOLATION OF HYALURONIC ACID FROM THE COCK'S COMB

Line 898 Use italics for (Raja clavata)

Lines 900- 901 Use italics for Mytilus galloprovincialis

Line 973 Add the authors to the citation: Robert G. Frykberg and Jaminelli Banks. Challenges in the Treatment of Chronic Wounds. Advances in Wound Care 2015, 4, (9), 560-582.

Lines 1012-1013 Use italics for Periplaneta Americana

Comment: The scientifically correct name is Periplaneta americana in italics, but is misspelled in the original title of the cited article.

Line 1082 Use italics for Myristica fragrans

Author Response

Comments and Suggestions for Authors

Title: Review Marine Biomaterials: Hyaluronan

Comment:

The authors present a review on Hyaluronan, a topic on which they demonstrate good expertise. However, papers are often cited as if they specifically dealt with the characteristics of biomaterials of marine origin while they do not mention them or mention them briefly, without underlining their importance and without proposing a comparison with biomaterials of different origins. Just to give just a few examples limited to the first few lines of the Introduction:

Line 23 The vast diversity of organisms living in the OCEANS presents a wealth of potential for developing high-value bioactive substances and biomaterials [1-3].

Line 24 Although many biomaterials have been derived from MARINE organisms in recent decades [4, 5],

Line 26 MARINE biomaterials have unique properties that make them promising materials for various biological and biomedical applications [6, 7]

Line 28 common MARINE polysaccharides and proteins, are important considerations in the development of these materials [8, 9].

Quickly reading the works cited, the papers [1-6, 8] do not seem to deal with either marine organisms or biomaterials of marine origin, except for occasional mentions of alginates.

Continuing this list is unnecessary, but I would suggest citing more relevant articles that underline the peculiarity of biopolymers of marine origin. I report a few below, but it will not be difficult for the authors to find others that are equally or more valid:

doi.org/10.3390/md18120589

doi.org/10.3390/md20060372

doi.org/10.3390/md20040219

doi.org/10.1021/acs.biomac.1c00013

doi.org/10.1016/j.isci.2023.106404

doi.org/10.1016/j.jece.2021.105895

doi.org/10.1016/j.marenvres.2016.03.007

doi.org/10.3390/polym13152482

Response: Thank you. All suggested references already mentioned in the text and more references related to marine sources have been added.

Line 3 Use superscript for a*

Response: Thank you. Corrected inside the text

Line 28 "are important considerations in the"

Do the authors mean important details (or aspects)?

Response: Thank you. Corrected inside the text

Line 37 I would not use a dedicated subparagraph for 1.1. Glycosaminoglycans

Response: Thank you. Corrected inside the text

Line 109 Figure 2 needs to be redesigned because only bivalve molluscs show the correct magnification

Response: Thank you. Corrected inside the text

Lines 212-214 Replace "For example, HA hydrogels can be used as injectable matrices to deliver cells or drugs directly to a specific tissue site[58].. For example, HA injections are commonly used in cosmetic procedures to" with "For example, HA hydrogels can be used as injectable matrices to deliver cells or drugs directly to a specific tissue site [58] and HA injections are commonly used in cosmetic procedures to" for avoid useless repetitions.

Response: Thank you. Corrected inside the text

Lines 294-297 The marine hyaluronan-based wound healing used in three typical wound repair types [74]. The marine hyaluronan-based wound healing used in three typical wound repair types is listed below in Table 2.

Response: Thank you. Corrected inside the text

Comment: Either the sentence was repeated unnecessarily or something is missing in the first sentence. Also in this case, the cited paper does not mention material of marine origin except as an incidental mention of alginates but without clarifying how they differ from other biomaterials of different origins.

Response: Thank you. Table 2 describe on effect of HA based on marine source molecular weight, chemical modification of HA, and preparation methods on treatments of different wound.

Line 307 Enlarge the firt box of Figure 4 to not separate the letter n from the word inflammation

Response: Thank you. Corrected inside the text.

Line 319 Replace "tissue repair. the inflammation" with "tissue repair. The inflammation"

Response: Thank you. Corrected inside the text.

Line 371 Use superscript for 2 in cm2

Response: Thank you. Corrected inside the text.

Line 374 Use subscript for 2 in (TcPO2)

Response: Thank you. Corrected inside the text.

Line 450 Use italics for in vitro

Response: Thank you. Corrected inside the text.

Line 463 Replace "Hussain et al [140]. published a study" with "Hussain and coworkers [140] published a study"

Response: Thank you. Corrected inside the text.

Line 471 Replace "Fahmy et al [141]. published a study" with "Fahmy and coworkers [141] published a study"

Response: Thank you. Corrected inside the text.

Line 480 Replace "Abdelrahman et al [142] have developed "with "Abdelrahman and coworkers [142] have developed"

Response: Thank you. Corrected inside the text.

Line 511 Use superscript for 2+ in (Ca2+)

Response: Thank you. Corrected inside the text.

Line 515 Replace the comma with a point in "most promising results [46], The"

Response: Thank you. Corrected inside the text.

Line 520 "Another type of HA-hydrogel is in the nervous system".

Comment: I don't understand what the authors mean by this sentence. Something is missing?

Response: Thank you. Corrected inside the text.

Line 522 Use italics for in vitro.

Response: Thank you. Corrected inside the text.

Line 522 Replace “[146, 147] [148].” with “[146-148].”

Response: Thank you. Corrected inside the text.

Line 530 Replace "the use of HA "with "The use of HA"

Response: Thank you. Corrected inside the text.

Line 532 Use italics for in vitro

Response: Thank you. Corrected inside the text.

Line 536 A hydrogel based on collagen I and hyaluronic acid (HA) was fabricated in a study,

Comment: I think it is appropriate to insert a citation here.

Response: Thank you. Corrected inside the text.

Line 553 "In another study, hydrogels were developed by combining alginate (ALG) and hya-"

Comment: If the relevant citation is [151], it must be inserted immediately, if it has been omitted, add it.

Response: Thank you. Corrected inside the text.

Line 567 Replace "Also, Studies have shown" with "Also, studies have shown"

Response: Thank you. Corrected inside the text.

Line 588 "Another injectable hydrogel based on marine hyaluronan derivatives" with "Another injectable hydrogel is based on marine hyaluronan derivatives"

Response: Thank you. Corrected inside the text.

Lines 652-655 "The effect of hyaluronan molecular weight on scaffold morphological properties shows promising results and the biocompatibility improved in presence of marine hyaluronan. The prepared materials are promising for tissue engineering application especially for skin healing applications[164]."

Comment: It does not appear that the hyaluron mentioned in this paper, [164], was of marine origin.

Line 680 Use italics for in vivo

Response: Thank you. Corrected inside the text.

Line 688 Replace "In Karine et al [169] utilized the electrospinning technique" with "Karine and coworkers [169] utilized the electrospinning technique"

Response: Thank you. Corrected inside the text.

Line 716 Use italics for in vitro

Response: Thank you. Corrected inside the text.

Lines 775-777 "These investigations should focus on evaluating the effects of the compounds when combined with different marine as a matrix."

Comment: The sentence seems incomplete. Did the authors intend to write marine biopolymers, marine biomaterials or something like that?

Line 779 Use italics for in vitro and in vivo

Response: Thank you. Corrected inside the text.

Line 780 Use italics for in vitro

Response: Thank you. Corrected inside the text.

Line 781 Use italics for in vivo

Response: Thank you. Corrected inside the text.

Line 783 Use italics for in vitro and in vivo

Response: Thank you. Corrected inside the text.

Line 752 The Future Trend and Beyond Limitations paragraph appears a bit repetitive. I think it could be summarized without much difficulty.

Response: Thank you. Corrected inside the text.

Lines 797-799 Abdel-Mohsen, A. M.; Jancar, J.; Abdel-Rahman, R. M.; Vojtek, L.; Hyršl, P.; Dušková, M.; Nejezchlebová, H., A novel in situ silver/hyaluronan bio-nanocomposite fabrics for wound and chronic ulcer dressing: In vitro and in vivo evaluations. International Journal of Pharmaceutics 2017, 520, (1), 241-253.

Comment: The first citation must be formatted in accordance with the rest of the bibliography. Also italics should be used for in situ, in vitro and in vivo.

Line 869-870 Use italics for (Lapemis curtus):

Response: Thank you. Corrected inside the text.

Line 891 Do not use capital letter for ISOLATION OF HYALURONIC ACID FROM THE COCK'S COMB

Response: Thank you. Corrected inside the text.

Line 898 Use italics for (Raja clavata)

Response: Thank you. Corrected inside the text.

Lines 900- 901 Use italics for Mytilus galloprovincialis

Response: Thank you. Corrected inside the text.

Line 973 ​Add the authors to the citation: Robert G. Frykberg and Jaminelli Banks. Challenges in the Treatment of Chronic Wounds. Advances in Wound Care 2015, 4, (9), 560-582.

Lines 1012-1013 Use italics for Periplaneta Americana

Response: Thank you. Corrected inside the text.

Comment: The scientifically correct name is Periplaneta americana in italics, but is misspelled in the original title of the cited article.

Response: Thank you. Corrected inside the text.

Line 1082 Use italics for Myristica fragrans

Response: Thank you. Corrected inside the text.

Round 2

Reviewer 1 Report

The authors have addressed all my concerns. I recommend it for publication after carefully checking the English.

Author Response

The authors have addressed all my concerns. I recommend it for publication after carefully checking the English.

Response. Thank you for the positive comment from the reviewer. The whole manuscript has already been corrected by a native English speaker.

Reviewer 3 Report

Lines 93-95 In 2010, the global HA treatment market was foreseen to generate over $13.5 billion in revenues, demonstrating an overall compound annual growth rate (CAGR) in excess of 8 .1%. The main sectors of extreme commercial interest are those related to dermal fillers,

Comment: I think it is appropriate to insert a citation here.

Line 98 2. Hyaluronic Acid

Comment: Format the paragraph title correctly.

Line 139 Figure 2

Comment: the figure 2 needs to be redesigned because only the image of bivalve molluscs show the correct magnification while the images of other taxa are too small in size.

Line 340 Unfraternally, all deacylated agents show disadvantages

Comment: perhaps did the authors mean “Unfortunately, all deacylated agents show disadvantages…”?

Line 357 Table 2

Replace vangiogenesis with angiogenesis;

Comment: for uniformity, always use uppercase or lowercase letters in the Table 2. For example replace Blend with blend, Chemically with chemically and so on, or vice versa replace wound healing with Wound healing and so on.

Line 388 Use bold for (Fig. 4).

Line 447 Use italics for in-situ.

Line 480 Use bold for (Fig. 4).

Line 481 Use italics for in-situ.

Line 495 Use italics for Escherichia coli

Line 495 Use italics for Staphylococcus aureus

Line 508 Use italics for in-vitro.

Line 540 Use italics for in situ.

Line 569 Use subscript for 2 in PEG-NH2

Lines 577-579 HA hydrogels have been used to encapsulate and deliver neural stem cells for the treatment of spinal cord injury and to support the growth and differentiation of neurons A hydrogel based on collagen I and hyaluronic acid (HA) in vitro[146-148].

Comment: perhaps did the authors mean: “Hydrogels based on collagen I and hyaluronic acid (HA) have been used to encapsulate and deliver neural stem cells for the treatment of spinal cord injury and to support the growth and differentiation of neurons in vitro [146-148].”?

Line 595 Replace {Xu, 2021 #135;Xu, 2021 #138;Zhang, 2020 #139} with [135, 138-139]

Lines 610-612 Replace “Another study developed hydrogels by combining alginate (ALG) and hyaluronan (HA) as a biofunctional platform for dermal wound repair. Also, Studies have shown [151].” with Another study developed hydrogels by combining alginate (ALG) and hyaluronan (HA) as a biofunctional platform for dermal wound repair [151].”

Comment: Hoping not to be indelicate, I would suggest that authors be very careful not to leave superfluous parts of the sentences after made the corrections requested by the reviewers, dedicating the necessary time to the revision and to an accurate re-reading of the whole text.

Line 639 Figure 6

Comment: the figure 6 is very blurry. The authors could download the high-res image (513KB) from https://www.sciencedirect.com/science/article/pii/S1385894721036184#f0005

Li, M.; Liang, Y.; Liang, Y.; Pan, G.; Guo, B., Injectable stretchable self-healing dual dynamic network hydrogel as adhesive anti-oxidant wound dressing for photothermal clearance of bacteria and promoting wound healing of MRSA infected motion wounds. Chemical Engineering Journal 2022, 427, 132039.

Line 649 Figure 7

Comment: also the figure 7 is very blurry. The authors could download the high-res image (660KB) from https://www.sciencedirect.com/science/article/pii/S0144861721009942#f0040

Yang, R.; Huang, J.; Zhang, W.; Xue, W.; Jiang, Y.; Li, S.; Wu, X.; Xu, H.; Ren, J.; Chi, B., Mechanoadaptive in-1241 jectable hydrogel based on poly(γ-glutamic acid) and hyaluronic acid regulates fibroblast migration for 1242 wound healing. Carbohydrate Polymers 2021, 273, 118607.

Line 711 Use bold for (Fig. 8).

Line 716 Gross view and microstructure of the silk fibroin/hyaluronan 3D scaffolds.

Comment: I don't understand why the authors use the opposing terms gross view and microstructure. The enlargements of all the images are the same so it is either a gross view or a microstructure. I would suggest that in this case it is the microstructure of the 3D scaffolds.

Line 775 Replace Solution with solution

Line 840 Use italics for in vitro.

Author Response

Lines 93-95 In 2010, the global HA treatment market was foreseen to generate over $13.5 billion in revenues, demonstrating an overall compound annual growth rate (CAGR) in excess of 8 .1%. The main sectors of extreme commercial interest are those related to dermal fillers,

Comment: I think it is appropriate to insert a citation here.

Response: Already done

Line 98 2. Hyaluronic Acid

Comment: Format the paragraph title correctly.

Response: Already done

Line 139 Figure 2

Comment: the figure 2 needs to be redesigned because only the image of bivalve molluscs show the correct magnification while the images of other taxa are too small in size.

Response: Already done

Line 340 Unfraternally, all deacylated agents show disadvantages

Comment: perhaps did the authors mean “Unfortunately, all deacylated agents show disadvantages…”?

Response: Already done

Line 357 Table 2

Replace vangiogenesis with angiogenesis;

Response: Already done

Comment: for uniformity, always use uppercase or lowercase letters in the Table 2. For example replace Blend with blend, Chemically with chemically and so on, or vice versa replace wound healing with Wound healing and so on.

Response: Already done

Line 388 Use bold for (Fig. 4).

Response: Already done

Line 447 Use italics for in-situ.

Response: Already done

Line 480 Use bold for (Fig. 4).

Response: Already done

Line 481 Use italics for in-situ.

Response: Already done

Line 495 Use italics for Escherichia coli

Response: Already done

Line 495 Use italics for Staphylococcus aureus

Response: Already done

Line 508 Use italics for in-vitro.

Response: Already done

Line 540 Use italics for in situ.

Response: Already done

Line 569 Use subscript for 2 in PEG-NH2

Response: Already done

Lines 577-579 HA hydrogels have been used to encapsulate and deliver neural stem cells for the treatment of spinal cord injury and to support the growth and differentiation of neurons A hydrogel based on collagen I and hyaluronic acid (HA) in vitro[146-148].

Response: Already done

Comment: perhaps did the authors mean: “Hydrogels based on collagen I and hyaluronic acid (HA) have been used to encapsulate and deliver neural stem cells for the treatment of spinal cord injury and to support the growth and differentiation of neurons in vitro [146-148].”?

Response: Already done

Line 595 Replace {Xu, 2021 #135;Xu, 2021 #138;Zhang, 2020 #139} with [135, 138-139]

Response: Already done

Lines 610-612 Replace “Another study developed hydrogels by combining alginate (ALG) and hyaluronan (HA) as a biofunctional platform for dermal wound repair. Also, Studies have shown [151].” with “Another study developed hydrogels by combining alginate (ALG) and hyaluronan (HA) as a biofunctional platform for dermal wound repair [151].”

Comment: Hoping not to be indelicate, I would suggest that authors be very careful not to leave superfluous parts of the sentences after made the corrections requested by the reviewers, dedicating the necessary time to the revision and to an accurate re-reading of the whole text.

Response: Already done

Line 639 Figure 6

Comment: the figure 6 is very blurry. The authors could download the high-res image (513KB) from https://www.sciencedirect.com/science/article/pii/S1385894721036184#f0005

Response: Already done

Li, M.; Liang, Y.; Liang, Y.; Pan, G.; Guo, B., Injectable stretchable self-healing dual dynamic network hydrogel as adhesive anti-oxidant wound dressing for photothermal clearance of bacteria and promoting wound healing of MRSA infected motion wounds. Chemical Engineering Journal 2022, 427, 132039.

Line 649 Figure 7

Comment: also the figure 7 is very blurry. The authors could download the high-res image (660KB) from https://www.sciencedirect.com/science/article/pii/S0144861721009942#f0040

Response: Already done

Yang, R.; Huang, J.; Zhang, W.; Xue, W.; Jiang, Y.; Li, S.; Wu, X.; Xu, H.; Ren, J.; Chi, B., Mechanoadaptive in-1241 jectable hydrogel based on poly(γ-glutamic acid) and hyaluronic acid regulates fibroblast migration for 1242 wound healing. Carbohydrate Polymers 2021, 273, 118607.

Response: Already done

Line 711 Use bold for (Fig. 8).       

Response: Already done

Line 716 Gross view and microstructure of the silk fibroin/hyaluronan 3D scaffolds.

Response: Already done

Comment: I don't understand why the authors use the opposing terms gross view and microstructure. The enlargements of all the images are the same so it is either a gross view or a microstructure. I would suggest that in this case it is the microstructure of the 3D scaffolds.

Response: Already done

Line 775 Replace Solution with solution

Response: Already done

Line 840 Use italics for in vitro.

Response: Already done